# Response of treatment-naive brain metastases to stereotactic radiosurgery

Chibawanye I. Ene [1,9] ✉, Christina Abi Faraj[1,9], Thomas H. Beckham[2], Jeffrey S. Weinberg [1], Clark R. Andersen[3], Ali S. Haider[1], Ganesh Rao[4], Sherise D. Ferguson[1], Christopher A. Alvarez-Brenkenridge[1], Betty Y. S. Kim [1], Amy B. Heimberger [5], Ian E. McCutcheon [1], Sujit S. Prabhu[1], Chenyang Michael Wang[2], Amol J. Ghia[2], Susan L. McGovern[2], Caroline Chung[2], Mary Frances McAleer[2], Martin C. Tom[2], Subha Perni[2], Todd A. Swanson[2], Debra N. Yeboa[2], Tina M. Briere[6], Jason T. Huse[7], Gregory N. Fuller [7], Frederick F. Lang [1], Jing Li[2], Dima Suki[1] & Raymond E. Sawaya[8]

With improvements in survival for patients with metastatic cancer, long-term local control of brain metastases has become an increasingly important clinical priority. While consensus guidelines recommend surgery followed by stereotactic radiosurgery (SRS) for lesions >3 cm, smaller lesions (≤3 cm) treated with SRS alone elicit variable responses. To determine factors influencing this variable response to SRS, we analyzed outcomes of brain metastases ≤3 cm diameter in patients with no prior systemic therapy treated with frame-based single-fraction SRS. Following SRS, 259 out of 1733 (15%) treated lesions demonstrated MRI findings concerning for local treatment failure (LTF), of which 202 /1733 (12%) demonstrated LTF and 54/1733 (3%) had an adverse radiation effect. Multivariate analysis demonstrated tumor size (>1.5 cm) and melanoma histology were associated with higher LTF rates. Our results demonstrate that brain metastases ≤3 cm are not uniformly responsive to SRS and suggest that prospective studies to evaluate the effect of SRS alone or in combination with surgery on brain metastases ≤3 cm matched by tumor size and histology are warranted. These studies will help establish multidisciplinary treatment guidelines that improve local control while minimizing radiation necrosis during treatment of brain metastasis ≤3 cm.

Brain metastasis remains a common manifestation of systemic cancer and remains a poor prognostic factor for cancer patients[1]. With improvement in overall survival of patients with brain metastasis being driven by more effective systemic treatment regimens, the incidence of brain metastasis has increased considerably in recent years[1]. While these new systemic therapies have demonstrated impressive clinical activity against primary cancers outside of the central nervous system (CNS), the blood-brain-barrier (BBB) limits the efficacy of systemic treatments within CNS[2], resulting in an unmet and urgent need for optimal local disease control strategies including surgery with stereotactic radiosurgery (SRS) or SRS alone in the management of brain metastases.

Several clinical trials have evaluated local management strategies for brain metastasis. The EORTC 22952–26001 study evaluated the impact of adjuvant whole brain radiation therapy (WBRT) after SRS or surgery for patients with 1–3 brain metastases[3]. Results showed that WBRT reduced intracranial relapses and neurological death but did not improve functional independence or overall survival. The RTOG

9508 study compared WBRT alone or WBRT followed by SRS boost for patients with 1–3 newly diagnosed brain metastases[4]. Results showed that WBRT and SRS boost improved functional outcomes and survival for patients with a single unresectable brain metastasis. An MD Anderson-led prospective trial compared observation to SRS after surgical resection of 1–3 brain metastasis with a post-operative cavity size that was <4 cm in maximum diameter[5]. Results showed that SRS of the surgical cavity in patients with complete resection of 1–3 brain metastasis significantly lowers local recurrence compared to post-operative observation. These results indicate that for patients presenting with 1–3 brain metastasis with large lesions >3 cm, surgery followed by SRS provides good functional outcomes and optimizes local disease control.

SRS alone is recommended for brain metastasis up to 3 cm maximum diameter (or 14 cm$^3$)[6]. Our previous study which analyzed local disease control rates for brain metastases in 135 patients ($N = 153$ lesions) who received SRS at MD Anderson between 1991 and 2001 demonstrated that the 1- and 2-year local control rates (LCRs) for tumors greater than 0.5 cm$^3$ (or 1 cm diameter) were significantly lower (56% and 24%, respectively) than for lesions smaller than 0.5 cm$^3$ (86% and 78%, respectively; $P = 0.0016$)[7]. These results indicate that lesions less than 0.5 cm$^3$ (or 1 cm diameter) are sensitive to SRS but lesions greater than 0.5 cm$^3$ (or 1 cm diameter) are less sensitive. The response of brain metastasis 1–3 cm maximum diameter to SRS is variable, likely due to other factors independent of size[8,9]. Therefore, the local management of brain metastasis ≤3 cm has evolved to incorporate other factors including cumulative tumor volume of intracranial disease (not limited to specific number of lesions), tumor location (eloquent v. non-eloquent) and primary tumor histology[10]. There are, however, no formal treatment guidelines that incorporate these factors into the management of patients with brain metastasis ≤3 cm.

Here, we retrospectively analyzed one of the largest single-institution cohorts of patients with 1–3 treatment-naïve brain metastasis who received framed SRS over a 25-year period. The purpose of this study was to identify variables or factors that influence the time to local treatment failure (TTF) and LCRs after SRS for brain metastasis in patients without prior or ongoing systemic treatment. These variables could then inform the design of prospective clinical trials seeking more effective radiotherapeutic and surgical treatment strategies that enhance local disease control and minimize variability in the outcomes of brain metastasis ≤3 cm treated with SRS.

## Results

### Demographics and treatments

Inclusion and exclusion criteria for patients and lesions includes in analysis are listed in Table 1. Specific criteria for determining response to SRS are show in in Table 2. Baseline patient characteristics, radiation treatment parameters and sequence of radiation therapy are listed in Tables 3 and 4. Among 1095 patients with 1733 lesions, 507 (46%) were female, and 588 (54%) were male. The median patient age was 62 years (range 16–95). Of 1095 patients, 616 had only one SRS-treated lesion, while 320 patients had two and 159 had three SRS-treated lesions. The most common primary tumor type was non-small cell lung cancer (36%), followed by melanoma (21%), breast cancer (12%), and renal cell carcinoma (6%). Seventy-four percent of lesions received single fraction Gamma Knife SRS with a mean periphery dose of 20 Gy and a range of 13.5–24 Gy. Twenty-six percent of lesions received single fraction LINAC SRS with a mean periphery dose of 18 Gy and a range of 8–22 Gy (one calvarial metastasis received 8 Gy). There were no fractionated treatments. Median target tumor diameter was 1.3 cm (range 0.28–2.96 cm).

### Outcomes following SRS for treatment-naïve brain metastasis.

Outcomes of the 1733 treatment-naïve lesions treated with SRS in eligible patients were analyzed (Fig. 1). Following SRS, 259 (15% of all

treated lesions) showed imaging findings concerning for LTF. Of these, 202 lesions (11% of all treated lesions) were deemed LTF based on specific criteria (Table 2). LTF was diagnosed after concerning radiographic findings led to surgical resection with pathology showing viable tumor only or mixed viable tumor and radiation necrosis (RN; $n = 110$ or 6% of all treated lesions) or clinical/radiographic signs necessitating a change in management ($n = 92$ or 5% of all treated lesions). ARE was identified in 57 lesions (4% of all treated lesions). Pure RN without viable tumor on pathology after surgery occurred in 22 lesions (1.3% of all treated lesions). 48 lesions (3% of treated lesions) had mixed pathology with both viable tumor and RN seen on pathology after surgery. Radiographic AREs that were deemed to be RN on ABTI and/or were responsive to steroids or Bevacizumab occurred in 18 lesions (1% of all treated lesions). In sum, radiographic and pathology proven RN occurred in 88 lesions (5% of all treated lesions). Hemorrhage or edema requiring surgical resection within 60 days of SRS occurred in 19 lesions (1% of all treated lesions). There were 26 patients with 36 lesions (2% of all treated lesions) with concerning imaging findings who were functionally not fit for further treatment or who chose not to proceed with further treatment. Perfusion data was available for 3 out of 36 lesions, all of which were consistent with a viable tumor signature. 17 out of the 26 patients with concerning imaging findings went on to hospice care after declining clinical intervention for the suspected intracranial progression.

### Univariate and multivariate analysis for factors influencing TTF.

Based on univariate analysis, year of SRS treatment ($P < 0.0001$), SRS dose ($P < 0.0001$), tumor size, primary tumor histology and SRS modality (LINAC v. GK SRS; $P < 0.0001$) significantly influenced TTF (Table 5). Multivariate analysis, however, showed that age, year of SRS, tumor size and primary tumor histology significantly influenced TTF (Table 5). The TTF ratio (ratio of TTF specified size range versus TTF of SRS susceptible lesions ≤0.5 cm; see methods for details) for lesions >0.5 and 1 cm, >1 and 1.5 cm, in diameter was shorter compared to lesions ≤0.5 cm, but this did not reach statistical significance (Table 5). However, the TTF ratios were significantly lower for lesions >1.5 and 2 cm (TTF ratio 0.31; 95% CI, 0.21–0.44; $P = 0.014$), >2 and 2.5 cm (TTF ratio 0.22; 95% CI, 0.16–0.32; $P = 0.0005$), >2.5 and 3 cm (TTF ratio 0.12; 95% CI, 0.07–0.20; $P = 0.0003$; Table 5). Multivariate analysis also showed that melanoma had a significantly shorter TTF relative to NSCLC and RCC (Table 5).

### Local control rates after SRS.

The 1- and 2- year local control rate (LCR) for all lesions treated with SRS were 82% and 78%, respectively with lesions censored at time of last imaging follow-up and at WBRT administration if it occurred before treatment failure of SRS treated lesion. The 1- and 2- year LCRs for lesions ≤0.5 cm were 93% and 90.5% respectively (Fig. 2 and Table 6). The 1-and 2-year LCRs for lesions in diameter ranges of 0.5–1, 1–1.5, 1.5–2, 2–2.5, 2.5–3 cm are 92.1 and 91%, 85.8 and 80.9%, 80.4 and 66.5%, 69.9 and 61.7% and 55.1 and 34.5% respectively (Fig. 2 and Table 6). Amongst tumor histologies, melanoma and breast cancers had a lower 2-year LCRs at 67.4% and 68.5% respectively (Table 7; Supplementary Fig. S1). Renal cell carcinoma (RCC) and non-small cell lung cancer (NSCLC) had a higher 2-year LCRs at 93.4% and 84.7% respectively (Table 7; Supplementary Fig. S1). LCR based on age and sex are shown in Supplementary Figs. S2 and S3.

### Illustrative cases.

We demonstrate the differential response of non-small cell lung and solitary melanoma brain metastatic lesions of similar sizes to GK SRS in an 83-year-old male patient with a history of NSCLC presenting with a solitary left parietal lesion of 1.42 cm diameter (or 1.5 cm$^3$) and a 62-year-old female patient with history of melanoma presenting with a solitary right frontal lesion of 1.45 cm diameter (or 1.6 cm$^3$), respectively (Fig. 3). Both lesions were treated with GK SRS at a dose of 20 Gy. At 1-year follow-up, the NSCLC lesion

**Table 1 | Inclusion and exclusion criteria**

| Inclusion criteria | Exclusion criteria |
|---|---|
| Patients with<br>• Newly diagnosed and untreated tumors undergoing framed LINAC or GK SRS between 6/1/1993 and 6/30/2018<br>• Maximum of 3 tumors treated with SRS | Patients with<br>• Prior surgical resection of the SRS treated lesion<br>• Prior cranial radiation therapy<br>• Prior or ongoing systemic treatments<br>• 4 or more tumors treated with SRS<br>• No postoperative imaging<br>• SRS treated lesions with concerning MRI findings but who were lost to follow-up<br>• SRS treated lesions >3 cm |

*LINAC* Linear Accelerator, *GK* Gamma Knife, *SRS* Stereotactic Radiosurgery, *MRI* Magnetic Resonance imaging.

**Table 2 | Specific criteria for response assessment following stereotactic radiosurgery for treatment-naïve brain metastases**

| Outcome | Criteria | Details |
|---|---|---|
| Local treatment failure (LTF) | Surgery and pathologic evaluation | Cases with sustained increase in lesion size followed by lesion resection with pathology showing tumor or mixed tumor and radiation treatment changes[a] |
| | Clinically and radiographically defined LTF | Combination of perfusion-weighted ABTI findings if available, increase in nodularity, hemorrhagic conversion, increased peri-tumoral edema, and/or sustained increase in lesion size on serial imaging<br>AND<br>resistant to steroids/bevacizumab therapy or received targeted therapy such as LITT and repeat SRS, or WBRT and systemic treatment to address local or concomitant local and distant treatment failure)[b] |
| Adverse radiation effect (ARE) | Surgery and pathologic evaluation | All cases with pure radiation necrosis on pathology |
| | Clinically and radiographically defined ARE | RN on ABTI if available, increase in lesion size/enhancement or peri-tumoral edema that eventually stabilized or resolved on subsequent imaging following treatment with steroids/bevacizumab |
| | Post-SRS hemorrhage or peri-tumoral edema requiring surgical resection[c] | No associated sustained increase in lesion size |

Cases with an associated increase in lesion size or perilesional edema that resolved spontaneously without any intervention were not deemed as LTF or ARE. The same SRS treated lesion was measured over several time points.
*ABTI* advanced brain tumor imaging, *LITT* laser interstitial thermal therapy, *SRS* stereotactic radiosurgery, *RN* radiation necrosis.
[a]If lesion resection was done within the first 60 days postoperatively and pathology showed viable tumor, such cases were classified as either LTF or ARE following multidisciplinary consensus.
[b]Patients who were not functionally fit for further treatment or opted to proceed without further treatment and were discharged for hospice care were considered local treatment failure.
[c]All surgeries in this category occurred within the first 60 days postoperatively and had tumor on pathology.

**Table 3 | Patient characteristics**

| Variable | | *N* | % |
|---|---|---|---|
| Patients | | | |
| | No. | 1095 | |
| | Total lesions | 1733 | |
| Age (years) | Median | 62 | |
| | Range | 16–95 | |
| Sex | Female | 507 | 46 |
| | Male | 588 | 54 |
| KPS | 30-60 | 22 | 2 |
| | 60-70 | 68 | 6 |
| | 80-90 | 646 | 59 |
| | 100 | 352 | 32 |
| | Median | 90 | |
| Primary | Non-small cell lung | 397 | 36 |
| | Melanoma | 225 | 21 |
| | Breast | 125 | 12 |
| | Renal | 67 | 6 |
| | Other | 281 | 26 |

showed a positive treatment response with near complete regression of the treated lesion. In contrast, the treated melanoma lesion showed treatment failure with an increase in lesion size at the 1-year follow-up.

## Discussion

The American Society of Clinical Oncology, Society for Neuro-Oncology and the American Society for Therapeutic Radiology and Oncology recommend single-fraction SRS for patients with brain metastases measuring 3 cm diameter (14 cm³)[6,11,12]. Fractionated SRS (e.g., 27 Gy in 3 fractions or 30 Gy in 5 fractions) is conditionally recommended for lesions ≥3 to 5 cm diameter (14–65 cm³)[12]. A meta-analysis of 24 SRS brain metastasis clinical trials showed that relative to single-fraction SRS, fractionated SRS reduces the risk of RN for lesions between 2–3 cm (4–14 cm³) but not for lesions >3 cm (>14 cm³)[13]. There was no significant difference in the 1-year local disease control between single- versus multifraction SRS for lesions over 2 cm. For lesions >4 cm diameter (>30 cm³), surgical resection is recommended followed by single fraction SRS. Based on these guidelines, many centers including the University of Texas MD Anderson Cancer Center perform single-fraction SRS for lesions up to 3 cm (14 cm³) and multi-fraction SRS for lesions >3 cm[14]. There remains significant variability, however, in the response of lesions 1–3 cm in diameter to single-fraction SRS, with some lesions demonstrating durable clinical responses and others failing treatment[7]. We hypothesized that to improve local control, all tumor intrinsic factors that significantly influence susceptibility of lesions to SRS need to be formally incorporated into the treatment guidelines for brain metastasis.

While prior studies have demonstrated the influence of tumor size on response to SRS[7,15], very few studies are powered to evaluate the influence of other tumor intrinsic factors on the response of treatment-naive lesions to SRS. This is an important consideration,

because evaluating SRS in this patient population would directly evaluate the biological response of brain metastases to SRS without the influence of previous or ongoing systemic treatments. To identify tumor intrinsic properties influencing response to SRS, we performed a retrospective analysis of SRS-treated brain metastases in patients

## Table 4 | Treatment characteristics

| Variable | | N | % |
|---|---|---|---|
| Year of stereotactic radiosurgery | | | |
| | Median | 2012 | |
| | Range | 1996–2018 | |
| Stereotactic radiosurgery modality | | | |
| | Gamma knife | 1281 | 74 |
| | Linear accelerator (LINAC) | 452 | 26 |
| Minimum peripheral stereotactic radiosurgery dose (Gy) | | | |
| | Gamma knife (to 50% Isodose) | | |
| | *Median* | 20 | |
| | *Range* | 13.5–24 | |
| | Linear accelerator; LINAC (to 81–95% Isodose) | | |
| | *Median* | 18 | |
| | *Range* | 8–22 | |
| Tumor diameter (cm) | | | |
| | Median | 1.3 | |
| | Range | 0.28–2.96 | |
| Diameter (cm) | Corr. volume (cm³) | | |
| (0.0, 0.5) | (0.0, 0.1) | 151 | 9 |
| (0.5, 1.0) | (0.1, 0.5) | 348 | 20 |
| (1.0, 1.5) | (0.5, 1.8) | 634 | 37 |
| (1.5, 2.0) | (1.8, 4.2) | 348 | 20 |
| (2.0, 2.5) | (4.2, 8.2) | 208 | 12 |
| (2.5, 3.0) | (8.2, 14.1) | 44 | 3 |
| Treatment sequence | | | |
| | Stereotactic radiosurgery only | 1565 | 90 |
| | Stereotactic radiosurgery + salvage whole brain radiotherapy | 168 | 10 |
| No. of treated brain metastases per patient | | | |
| | 1 | 616 | 56 |
| | 2 | 320 | 29 |
| | 3 | 159 | 15 |

with treatment-naïve brain lesions. We find that in treatment-naïve brain lesions that 3 cm, tumor size and primary histology significantly influence long term local control after SRS as initial local therapy. This indicates that the current recommendation for treatment of brain metastases based solely on size is not sufficient to identify lesions or patients who will respond best to treatment. Consistent with our prior report[7], lesions 1.5 cm demonstrated higher 2-year LCRs (over 81%), while lesions over 1.5 cm had significantly lower 2-year LCRs (66.5%). Even within the former group (lesions 1.5 cm), histology plays a role in the susceptibility to SRS as exemplified in the case illustration comparing the response of melanoma to NSCLC brain metastasis and in preclinical predictive models validated in tumor cell lines[16].

Predictive models using cell lines and in vivo data have indicated that multifraction SRS may be equally effective for local control when compared with single fraction SRS for radioresistant lesions[16]. One study found that for melanoma brain metastasis, three 8-Gy SRS fractions (EQD2) would have similar tumor control as that of a single fraction 20 Gy SRS[16]. At MD Anderson, single fraction SRS dosing is modified based on tumor volume in accordance with cavity volumes outlined in N107C[17]. For lesions between 1.5–3 cm, treatment is not uniform with some cases receiving single fraction SRS and others multifraction SRS. For lesions larger than 3 cm in patients who are not good surgical candidates, fractionation is typical, most often with 27 Gy in 3 fractions. In larger tumors, multifraction treatments are done to minimize the risk of toxicity or RN[18]. It is unclear, however, whether multifraction SRS can achieve local control in radioresistant histologies between 1.5–3 cm while limiting the incidence of RN. The SAFESTEREO study is an ongoing phase II prospective and randomized study that is comparing the incidence of adverse local events in patients with brain metastasis treated with 1 or 3 fractions versus 5 fractions (NCT05346367)[19]. Results from this prospective study will hopefully elucidate whether multifraction SRS could achieve effective local control in radioresistant histologies between 1.5–3 cm (identified in our study as the vulnerable range), with decreased incidence of RN. Another prospective study that remains in the concept phase is a histology-specific prospective trial comparing single fraction SRS to multifraction SRS (9 Gy × 3 fractions; NRG Oncology). Here, patients with radioresistant histologies like melanoma who have lesions between 1.5 cm and 3 cm will be randomized to receive single fraction SRS or multifraction SRS. The primary outcomes are incidence of RN and LTF. Altogether, these trials will identify radiotherapy-based approaches that improve local control while minimizing RN in radioresistant histologies between 1.5–3 cm.

A second approach to improve local control for radioresistant histologies is surgical resection combined with RT or SRS. Currently,

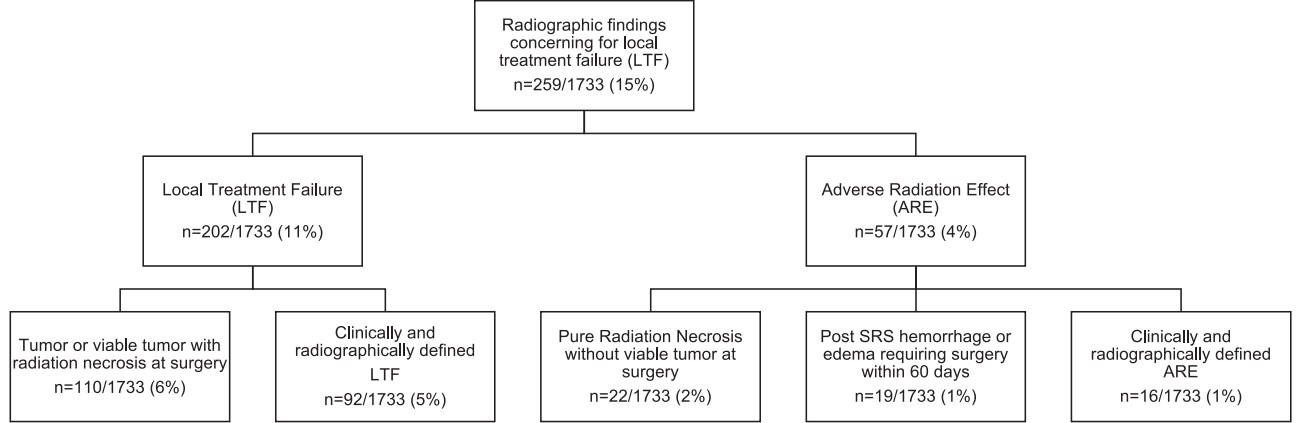

**Fig. 1 | Lesion outcomes following stereotactic radiosurgery for treatment-naïve brain metastases.** Outcomes following Stereotactic Radiosurgery (SRS) for treatment-naïve brain metastasis. SRS treated lesions that met inclusion criteria (n = 1733) were categorized as Local Treatment Failure (LTF) or Adverse Radiation Effect (ARE) according to specific criteria in Table 2. Source data provided as a Source data file.

**Table 5 | Univariate and multivariate analysis**

| Tumor control | Time to treatment failure ratio (95% CI) | p-value | Patient (n) | Lesions (n) | Local treatment failures (n) |
|---|---|---|---|---|---|
| Univariate analysis | | | | | |
| Age | 0.992 (0.977–1.007) | 0.3 | 1095 | 1733 | 202 |
| Sex (Female v. Male) | 0.716 (0.489–1.048) | 0.09 | 1095 | 1733 | 202 |
| Year of SRS | 1.135 (1.095–1.178) | <0.0001*** | 1095 | 1733 | 202 |
| KPS | 1.01 (0.992–1.03) | 0.28 | 1073 | 1696 | 202 |
| SRS Dose | 1.299 (1.179–1.431) | <0.0001*** | 1095 | 1733 | 202 |
| Tumor diameter v. (0.0, 0.5) | | | | | |
| (0.5, 1) | 1.111 (0.445–2.776) | 1 | 1095 | 1733 | 202 |
| (1, 1.5) | 0.429 (0.204–0.898) | 0.22 | | 1733 | 202 |
| (1.5, 2) | 0.25 (0.114–0.551) | 0.008** | | | |
| (2, 2.5) | 0.221 (0.096–0.466) | 0.002** | | | |
| (2.5, 3.0) | 0.1 (0.037–0.271) | <0.0001*** | | | |
| Primary tumor | | | | | |
| Melanoma v. NSCLC | 0.427 (0.256–0.713) | 0.010* | 1095 | 1733 | 202 |
| Breast v. NSCLC | 0.57 (0.354–0.974) | 0.24 | | | |
| Breast v. Melanoma | 1.335 (0.749–2.38) | 0.86 | | | |
| RCC v. NSCLC | 1.861 (0.783–4.421) | 0.62 | | | |
| RCC v Melanoma | 4.357 (1.806–10.512) | 0.009* | | | |
| RCC v. Breast | 3.264 (1.334–7.987) | 0.07 | | | |
| SRS modality | | | | | |
| Framed LINAC v. GammaKnife SRS | 0.329 (0.224–0.483) | <0.0001*** | 1095 | 1733 | 202 |
| Multivariate analysis | | | | | |
| Age | 0.988 (0.972–0.999) | 0.04* | | | |
| Year of SRS | 1.123 (1.084–1.163) | <0.0001*** | | | |
| Tumor diameter v. (0.0, 0.5) | | | | | |
| (0.5, 1) | 1.34 (0.89–2.03) | 0.98 | | | |
| (1, 1.5) | 0.54 (0.39–0.77) | 0.48 | | | |
| (1.5, 2) | 0.31 (0.21–0.44) | 0.014* | | | |
| (2, 2.5) | 0.22 (0.16–0.32) | 0.0005** | | | |
| (2.5, 3.0) | 0.12 (0.07–0.20) | 0.0003*** | | | |
| Primary tumor | | | | | |
| Melanoma v. NSCLC | 0.36 (0.28–0.45) | 0.0001*** | | | |
| Breast v. NSCLC | 0.53 (0.41–0.69) | 0.09 | | | |
| Breast v. Melanoma | 1.49 (1.13–1.96) | 0.59 | | | |
| RCC v. NSCLC | 1.41 (0.88–2.25) | 0.95 | | | |
| RCC v Melanoma | 3.94 (2.43–6.37) | 0.036* | | | |
| RCC v. Breast | 2.64 (1.62–4.32) | 0.28 | | | |

Time to treatment failure (TTF) model-adjusted differences as contrasts among the levels of the discrete variables, with two-sided Tukey-adjusted p-values. Source data and exact p values are provided as a Source Data file.
*CI* Confidence interval, *LINAC* Linear Accelerator, *GK* Gamma Knife, *SRS* Stereotactic Radiosurgery, *MRI* Magnetic Resonance imaging, *KPS* Karnofsky Performance Score, *NSCLC* Non-small cell lung cancer, *RCC* Renal cell carcinoma.
*p < 0.05; **p < 0.005; ***p < 0.0005.

there is limited consensus on the need for surgery for lesions ≤3 cm maximum diameter and hence there is significant variability in the management of lesions within this range with a propensity to perform surgery for larger lesions especially if associated with symptomatic mass effect and/or edema[20]. Therefore, a prospective randomized clinical trial is needed to determine whether there is a role for surgery in the management of radioresistant histologies that are 3 cm. In this trial, patients with radioresistant histologies between 1.5 cm and 3 cm are randomized to either surgery followed by single fraction or multifraction SRS to the cavity or single fraction SRS-only to the lesion (standard of care for lesions <3 cm). Another ongoing approach that could uncover the role of surgery for management of radioresistant brain metastasis is surgery followed by Cs-131 collagen tile brachytherapy. In a phase 1/2 human clinical trial, post-operative Cs-131 collagen tile brachytherapy for newly-diagnosed brain metastasis

(n = 24 patients and histologies including lung, renal, melanoma and cervical) was associated with no local recurrences or RN[21]. At MD Anderson, there is an ongoing phase 3 randomized controlled trial of post-surgical stereotactic radiotherapy (SRT, standard of care) versus surgically targeted radiation therapy (STaRT) with Cs-131 collagen tile brachytherapy for treatment of newly-diagnosed metastatic brain tumors or ROADS (Radiation One and Done Study; NCT 04365374). We anticipate that data from this study will also provide insights into whether Cs-131 collagen tile brachytherapy following surgical resection may provide more effective local control while minimizing RN especially for radioresistant histologies. Pre-operative SRS is another strategy that may improve local control while minimizing RN and incidence of leptomeningeal disease (LMD). To this end, there are several ongoing phase III randomized studies including the NRG-BN012 (NCT05438212) and an MD Anderson study that are comparing

preoperative SRS to postoperative SRS for newly-diagnosed brain metastasis[22]. We expect that these ongoing prospective studies will provide a framework for designing future prospective studies focused on identifying the optimal synergistic approach using surgery and radiotherapy for radioresistant histologies between 1.5–3 cm.

There are also systemic avenues to enhance response of brain metastasis to RT. Administration of radiosensitizers have also been proposed to overcome the intrinsic resistance mechanisms of brain metastasis to RT[23]. It has been shown that in preclinical models, competitive inhibitors of the DNA Damage repair (DDR) genes ATM and ATR Kinase sensitizes NSCLC brain metastasis to radiation therapy[24]. Histology (squamous cell carcinoma or adenocarcinoma) influenced the types of DDR alterations in NSCLC indicating that histology may also play a role in determining the susceptibility of NSCLC to radiosensitizers. Following pre-clinical studies, we anticipate that

human clinical trials will assess the role of radiosensitizers in the response of radioresistant histologies to SRS. There are reports of synergy between RT and systemically administered immune check point inhibitors (ICI) in preclinical and clinical studies for primary and metastatic brain tumors[25–27]. This approach with ICIs leverages the potential immunostimulatory effect of RT on the tumor micro-environment[28]. In NSCLC, the addition of ICI to RT boosts the infiltration of anti-tumor immune cells which enhance local control[27]. The synergy between RT and ICI, however, also appears to be histology-specific and ICI-specific indicating that tumor-specific micro-environment factors may influence susceptibility to these treatments[26,29]. The use of RT with ICI, may increase the risk of RN[30,31]. In patients with melanoma brain metastasis treated with SRS and anti-CTLA-4 and/or anti-PD-1 at MD Anderson Cancer Center, a multivariate analysis showed that use of chemotherapy within 6 months of SRS and number of lesions treated were predictive of increased RN risk (HR 2.20, 95% CI 1.22–3.97, $p = 0.009$; HR 1.09, 95% CI 1.03–1.15, $p = 0.002$)[31]. Ultimately, prospective randomized clinical trials are needed to determine the influence of tumor histology, prior treatments and number of lesions

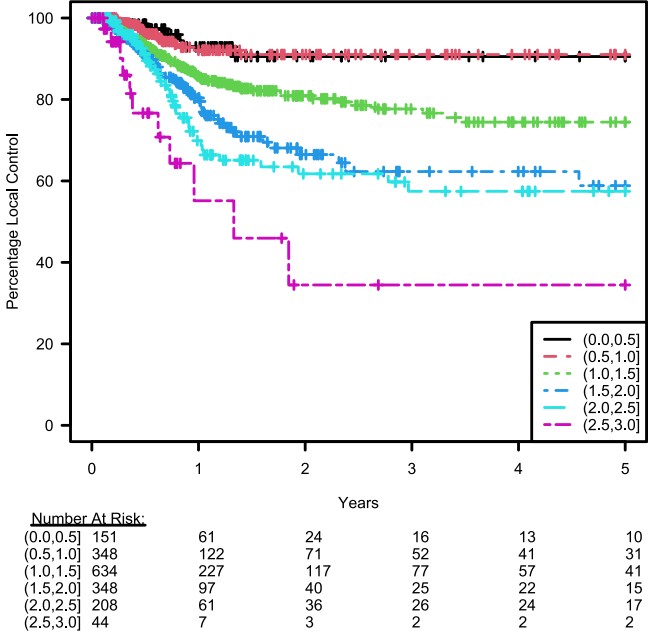

**Number At Risk:**

| | | | | | |
|---|---|---|---|---|---|
| (0.0,0.5] | 151 | 61 | 24 | 16 | 13 | 10 |
| (0.5,1.0] | 348 | 122 | 71 | 52 | 41 | 31 |
| (1.0,1.5] | 634 | 227 | 117 | 77 | 57 | 41 |
| (1.5,2.0] | 348 | 97 | 40 | 25 | 22 | 15 |
| (2.0,2.5] | 208 | 61 | 36 | 26 | 24 | 17 |
| (2.5,3.0] | 44 | 7 | 3 | 2 | 2 | 2 |

**Fig. 2 | Local control after SRS based on tumor diameter.** Kaplan–Meier curves showing percent local control of brain metastasis lesions over time after Stereotactic Radiosurgery (SRS) stratified by diameter (cm). *x-axis is censored at 5 years post-SRS as only 5 lesions under surveillance failed SRS after 5 years.* Source data provided as a Source data file.

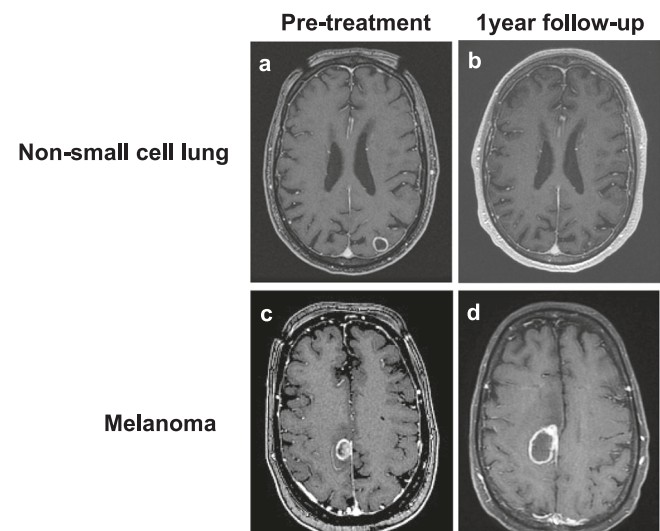

**Fig. 3 | Case Illustrations. Differential response of similar size brain metastasis from distinct primaries to Stereotactic Radiosurgery (SRS).** T1-weighted MRI with contrast showing pre-treatment and 1-year follow-up SRS treatment scans for 1.42 cm (or 1.5 cm³) on-small cell lung cancer (**a** pretreatment, **b** follow-up) and 1.45 cm (1.6 cm³) melanoma (**c** pretreatment, **d** follow-up).

## Table 6 | 1- and 2- year % local control (LC) based on tumor size

| Diameter (cm) | (0–0.5] | (0.5, 1] | (1, 1.5] | (1.5, 2] | (2, 2.5] | (2.5, 3] |
|---|---|---|---|---|---|---|
| 1-year % LC (CI95Min–CI95Max) | 92.9 (87.4–98.8) | 92.1 (88.1–96.2) | 85.8 (82.2–89.5) | 80.4 (74.7–86.5) | 69.9 (61.6–79.3) | 55.1 (35.3–86.2) |
| 2-year % LC (CI95Min–CI95Max) | 90.5 (83.6–98) | 91 (86.6–95.7) | 80.9 (76.5–85.6) | 66.5 (58.4–75.7) | 61.7 (52.5–72.6) | 34.5 (15.4–77.1) |

Predictions from Stratified Kaplan–Meier Summaries. Source data provided as a Source data file.
*LC* Local control, *CI* Confidence interval.

## Table 7 | 1- and 2- year % local control (LC) based on tumor histology

| Primary tumor | RCC | NSCLC | Breast | Melanoma | Other |
|---|---|---|---|---|---|
| 1-year % LC (CI95Min–CI95Max) | 96.4 (91.4–100) | 88.9 (85.6–92.2) | 82 .6 (76.4–89.3) | 73.4 (67.2–80.1) | 83.9 (78.8–89.3) |
| 2-year % LC (CI95Min–CI95Max) | 93.4 (86.1–100) | 84.7 (80.5–89.1) | 68.5 (59.6–78.7) | 67.4 (60.5–75.2) | 78.6 (72.4–85.4) |

Predictions from Stratified Kaplan–Meier Summaries. Source data provided as a Source data file.
*LC* Local control, *CI* Confidence interval, *RCC* Renal cell carcinoma, *NSCLC* Non-small cell lung cancer.

treated on the response (Local control v. RN) of brain metastasis with radioresistant histologies to SRS and ICI.

Distinguishing RN (or pseudoprogression) from true local tumor progression without a pathologic evaluation poses a significant challenge, as both can present with similar clinical features and radiographic findings[32]. Imaging technology is still in the early stages of reliably differentiating RN from true local progression[33]. Additionally, the presence of viable tumor on pathologic evaluation following early surgical resection (within 60 days post-SRS) may not reliably indicate treatment failure in cases such as hemorrhagic conversion. We addressed this challenge by diagnosing LTF based on a combination of radiographic findings, clinical management, and extensive patient follow up and outcome assessment. We also utilized perfusion weighted ABTI, when available, to distinguish RN from true progression (Supplementary Fig. S4). In prospective clinical trials, LITT was shown to be effective for symptomatic RN and allowed for a decreased dependency on steroids[34]. Therefore, we performed LITT for symptomatic RN to minimize complications from long term steroids or bevacizumab use (Supplementary Fig. S4). In the case of early salvage resection showing mixed tumor and RN on pathology, we differentiated local progression from ARE after a multidisciplinary characterization of radiographic features and the presence of sustained tumor progression despite the administration of steroids or bevacizumab. The 5% overall incidence of RN across all treated lesions (pure, mixed, clinical and radiographic; see Table 2) in this study is consistent with prior reports of a RN rate of 5–26% per lesion[35–37]. Additionally, SRS is associated with a lower incidence of RN in tumors <2 cm[37] and over 85% of the treated lesions in this study were <2 cm. These results support the relatively lower RN rate noted in this study.

From a clinical perspective, although size, histology and location are part of the discussion during management of patients with brain metastasis[38], there are currently no standardized or high level guidelines that incorporate these factors a decision-making algorithm that weights risk of local failure with risk of RN. Until prospective studies discussed above are completed, we are developing an MD Anderson brain metastasis nomogram that allows for determination of risk of local failure and RN based on a combination of factors including tumor size and histology. This will allow for better patient selection for treatment with SRS or surgery with SRS. Our hope is that as the overall survival of patients with metastatic cancer continues to improve, refined selection of the most appropriate local therapy approach for brain metastasis will improve outcomes and the quality of life of patients with brain metastases.

## Study limitations

This is a single institution study that is subject to selection bias. It is also a retrospective study, and our findings need to be validated in prospective and randomized clinical trials. Although there are some minor differences between findings from our study and prior studies assessing the susceptibilities of brain metastasis to SRS, the main findings from our study regarding the radioresistance of melanoma are consistent with published literature. One consideration that may contribute to the differences between our study and prior studies may lie in the fact that our cohort is exclusively treatment-naïve while other studies pool lesions from patients receiving a diverse array of local or systemic treatments. Our multivariate analysis also showed that year of treatment influenced TTF. Therefore, it is possible that other factors such as the availability of more effective brain penetrant agents and immunotherapy administered after SRS may have improved outcomes in the post-2009 era. As our retrospective analysis does not include frameless SRS cases, it is unclear if a frameless approach influences the results described in the study. Finally, despite our thorough criteria characterizing local tumor progression, it remains challenging to reliably differentiate pseudoprogression from tumor progression in certain cases especially in the patients with suspected intracranial

progression based on MRI findings that ultimately were not functionally fit for further treatment or declined clinical intervention, although these cases represented a small fraction of treated lesions in our cohort (2% of all treated lesions).

In summary, treatment-naive brain metastases that are ≤3 cm diameter or 14 cm³ (recommended cut-off for single-fraction SRS) are not uniformly responsive to SRS. Within this group of brain metastases, we find that tumor size (>1.5 cm diameter or 1.8 cm³) and primary histology (melanoma) have significantly shorter TTF and consequently lower 1- and 2-year local control rates after SRS relative to lesions <1.5 cm. Since our results are not influenced by prior or ongoing systemic treatments, these findings indicate that tumor intrinsic properties significantly influence susceptibility to SRS. To establish standardized and multidisciplinary clinical guidelines regarding the optimal brain metastatic lesion size and histologic criteria that portend a favorable LC and low risk of RN after SRS or surgery with SRS, prospective clinical trials enrolling patients matched by tumor size and histology are warranted.

## Methods

### Study design and patient characteristics

The Institutional Review Board (IRB) at the University of Texas MD Anderson Cancer Center approved this retrospective study and a waiver of informed consent, which included a chart review of 3000 patients with metastatic brain lesions treated with the frame-based LINAC and Gamma Knife (GK) SRS from June 1, 1993 to June 30, 2018. Patients with a maximum of 3 treatment-naive lesions were included (Table 1). We included patients with prior surgical resection of a different brain metastasis. We excluded patients who had whole brain radiation therapy (WBRT) and/or chemotherapy prior to SRS and those with no postoperative imaging. We excluded lesions that were >3 cm in maximum dimension (Table 1). LINAC SRS prescription doses were delivered to the 81–95% isodose line while GK SRS prescription doses were delivered to the 50% isodose line, per our standard practice at MD Anderson Cancer Center during the study time period[7,39]. Local treatment failure (LTF) was defined either radiographically or by pathology after surgery for radiographically suspected LTF (Table 2). Radiographic progression of an SRS-treated lesion was defined as a sustained increase in tumor size on serial imaging, or, when available, increased perfusion suggestive of viable tumor on advanced brain tumor imaging (ABTI). Such changes were resistant to treatment with steroids or bevacizumab and required a change in the clinical management, including WBRT for local and distant recurrence, systemic treatment, or targeted therapy (repeat SRS or laser interstitial thermal therapy (LITT)). To distinguish tumor progression from pseudoprogression, perfusion data was reviewed from the ABTI if available. Patients with concerning intracranial imaging findings, who did not have an ABTI but were offered further treatment such as WBRT and chose not to proceed with further treatment or had poor functional status were deemed as having LTF based on the clinical decision to offer further treatment for a suspected LTF (Fig. 1; Pseudoprogression cannot be ruled out entirely in patients who chose not to proceed with treatment). For patients who underwent salvage surgical resection, pathology showing viable tumor was deemed LTF. Adverse radiation effects (ARE) were documented and included clinically or radiographically defined ARE requiring steroids or bevacizumab, decreased perfusion suggestive of radiation necrosis (RN) on ABTI when available, pure RN on pathology following surgical resection, and post-SRS hemorrhage or peri-tumoral edema requiring surgical resection within the first 60 days postoperatively. Patients with concerning imaging findings who did not have a follow-up clinical encounter to address such changes were excluded. Cases with concerning imaging findings that resolved spontaneously without intervention were not deemed as LTF or ARE.

Patient characteristics, radiation treatment parameters and sequence of radiation therapy are listed in Tables 3 and 4. Follow-up data was obtained for 1095 patients and censored at last imaging follow-up or ARE. At the time of analysis, the mean duration of follow-up from time of SRS was 19 months for all patients. Follow-up imaging was obtained at the discretion of the team treating the patient (neurosurgeon, medical oncologist, and radiation oncologist). Since the goal of the project was to identify LTF, we did not include distant failure as an endpoint.

## Statistical analysis

TTF was summarized by Kaplan–Meier method for discrete variables, with each lesion treated as an independent sample (within-patient clustering was subsequently accommodated within appropriate models)[40]. Follow-up was censored at last imaging follow-up or ARE. Since additional radiation therapy to the brain via WBRT may alter the outcomes following SRS, patients were censored at time of post-SRS WBRT if local failure of the SRS treated lesion had not occurred when WBRT was given. TTF was modeled by accelerated failure time models with log-logistic distribution selected per Akaike Information Criteria among Weibull, hat exponential, Gaussian, logistic, log-normal, and log-logistic distributions, and verified by residual plot overlaid on the distribution as well as deviance residual plots for covariates, and clustering on patients to control for repeated events. Rather than the reported hazard ratios typical of Cox models, accelerated failure time models report the TTF ratio which compares the TTF across the size ranges (versus 0.5 cm) or across multiple histologies. The TTF ratio is approximately interpretable as the inverse of hazard ratios. A multi-variable accelerated failure time model of TTF was found by exhaustive variable selection by comparing the Akaike Information Criterion of models of all combinations of variables (age, year of treatment, primary tumor histology, discrete diameter, LINAC vs. Gamma knife SRS, KPS, SRS dose). The resulting optimal model assessed the association between TTF and the variables of age, year of treatment, primary tumor histology, discrete diameter, and post-SRS WBRT status. Model-adjusted differences among the levels of discrete variables in TTF were assessed by Tukey test. Age, Sex, year of treatment, primary tumor histology, tumor size, SRS modality (LINAC vs. GK SRS), pre-SRS KPS, and SRS dose were assessed in a univariate analysis. Statistical analyses were performed using R statistical software[41]. All statistical tests utilized two-sided alpha = 0.05 for a 95% level of statistical confidence. Survival modeling was performed using the "survival" package[42,43]. Assessment of differences among discrete variable levels in the accelerated failure time model were estimated using the emmeans package which includes adjusted means weighted proportionally to covariate marginal frequencies[44].

## Reporting summary

Further information on research design is available in the Nature Portfolio Reporting Summary linked to this article.

## Data availability

Although de-identified source data are provided with this paper, the complete clinical data that support the findings of this study are not openly available given IRB restrictions on human clinical data. Anonymized data are available from the corresponding author upon reasonable request and IRB approval. Following IRB approval, the de-identified clinical data will be made available within 2 weeks. Source data are provided with this paper.

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

## Acknowledgements

C.I.E. is supported by the MD Anderson Cancer Center Physician Scientist Program and the National Cancer Institute/ National Institutes of Health, Early Surgeon Scientist Program. We thank Preeti Ramadoss, Ph.D., for critical review and editing of the manuscript.

## Author contributions

C.I.E., D.S. and R.E.S. conceptualized the study. C.I.E., C.A.F., A.H. performed chart review. C.R.A. was the statistician. R.E.S., G.R., S.D.F., A.B.H., I.E.M., J.S.W., S.S.P., F.F.L., D.N.Y., J.L., T.H.B., B.Y.S.K., S.L.M., M.F.M., T.M.B., performed stereotactic radiosurgery cases. C.A.B., C.M.W., A.J.G., C.C., M.C.T., S.P., T.A.S. reviewed and edited the manuscript. J.T.H. and G.N.F. performed pathological analysis on tissue from surgery. C.I.E., C.A.F., T.B, D.S. and R.E.S. analyzed and interpreted the results. C.I.E., D.S., T.B., R.E.S. supervised the project. C.I.E., C.A.F., R.E.S. wrote the manuscript. All authors reviewed and edited the work. All authors have read and agreed to the submitted version of the manuscript.

## Competing interests

The authors declare no competing interests.

## Additional information

[1]Department of Neurosurgery, The University of Texas M D Anderson Cancer Center, Houston, TX, USA. [2]Department of Radiation Oncology, The University of Texas M D Anderson Cancer Center, Houston, TX, USA. [3]Department of Biostatistics, The University of Texas M D Anderson Cancer Center, Houston, TX, USA. [4]Department of Neurosurgery, Baylor College of Medicine, Houston, TX, USA. [5]Department of Neurological Surgery, Malnati Brain Tumor Institute of the Lurie Comprehensive Cancer Center, Feinberg School of Medicine, Northwestern University, Chicago, IL, USA. [6]Department of Radiation Physics, Division of Radiation Oncology, The University of Texas M D Anderson Cancer Center, Houston, TX, USA. [7]Department of Pathology, The University of Texas M D Anderson Cancer Center, Houston, TX, USA. [8]Faculty of Medicine and Medical Affairs, American University of Beirut, Beirut, Lebanon. [9]These authors contributed equally: Chibawanye I. Ene, Christina Abi Faraj. ✉e-mail: cene@mdanderson.org

