## [Peer Review File · Nature Communications]

REVIEWER COMMENTS

Reviewer #1 (Remarks to the Author):

Review of the Manuscript by Ene and colleagues

Introduction: Ene and colleagues investigate the efficacy of stereotactic radiosurgery (SRS) on “smaller” brain metastases (≤ 3 cm in diameter) in patients with no prior systemic therapy. Their study is rooted in the observation that while SRS alone is the treatment of choice for lesions ≤ 3 cm, the responses vary. This study aims to discern the factors behind this variability.

Key results: A significant number of lesions showed MRI findings consistent with local treatment failure (LTF) post-SRS. Notably, tumor size (especially > 2 cm) and melanoma histology were associated with higher LTF rates. Their assertion that brain metastases ≤ 3 cm do not uniformly respond to SRS alone and that guidelines integrating tumor size and histology are required is quite contentious.

Validity: The manuscript offers valuable insights into SRS's variable effectiveness based on tumor size and histology. However, the recommendation against SRS for certain tumor sizes and histologies seems premature, especially given the well-established efficacy and minimal side effects of SRS. Data are reported during a 25 years time frame with GK + Linac based SRS – prescription differences? I assume this was largely different, but is of utmost importance when interpreting LC data; the following statement needs to be discussed further:

“Patients with concerning imaging findings who were functionally not fit for further treatment or who chose not to proceed with further treatment were also deemed as having local treatment failure” – problematic as there may have been several cases of pseudoprogression; outcome of these patients?

Significance: While the article points out the potential radioresistance of melanomas and the size-dependent response to SRS, these findings are not novel. However, the data's significance lies in its comprehensive analysis from a size perspective, which is often overlooked in pooled data.

Data and methodology: The retrospective analysis of a large single-institution cohort over 25 years is a significant strength. However, there are inconsistencies, which may still arise from misallocation/misinterpretation.

Analytical approach: While the statistical approach, particularly concerning size, is rigorous and differentiates this study from others, some results contradict established literature, raising questions about potential errors or biases. In the methods section, the paragraph could be shortened and simplified a bit.

Suggested improvements:

- please correct line 96 regarding trial EORTC 22952-26001.
- please clarify: "The majority of the lesions were treated with Gamma Knife SRS (74%) and the median dose was 20 Gy (range 8-24)." – means there were hypofractionated courses included as well?
- Please discuss the main flaw: Radiation necrosis (RN) based on 230 pathology after surgery occurred in 28 lesions (2% of all treated lesions) – far too low compared to expectations when delivering ablative doses
- Information on margins used, Dx values and when hypofractionation was used are largely missing
- Please discuss the alternative scenario that hypofractionation might overcome existing limitations – as even surgery + SRS is associated with high failure rates, specifically when the cavity volume was large

Clarity and context: The article is generally clear, but it could benefit from a more on depth discussion contextualizing its findings within the larger body of literature. It is vital to recognize the novel contributions without overstating them.

Overall: The manuscript brings valuable insights, especially with its rigorous statistical analysis concerning tumor size. However, it is crucial to reconcile findings with existing literature and not draw broad conclusions without the robustness of randomized trials. Specifically as due to Kjellberg's observation, prescription is lowered when size grows as RN shall be avoided - which in turn explains lower control rates if the RN rate is low.

Reviewer #2 (Remarks to the Author):

This is the biostatistical review. This is a retrospective analysis of data from a single center including patients from 25 years.

The authors perform a lesion-based analysis with endpoint time to local failure. They fit an AFT model with log-logistic distribution accounting for clustered data due to multiple lesions per patients. While the

overall approach is fine, there are some discrepancies between described methods and reported results, and some points need clarification.

Endpoint TTF: the authors state that FU was censored at last FU, salvage surgical resection, or ARE. But they also state 'For patients who underwent salvage surgical resection, pathology showing viable tumor was deemed LTF'. Please clarify.

How were the other lesions of the same patient treated in case one lesion showed local failure?

Endpoint OS: I do not think a lesion-based analysis for OS is sensible (suppl fig 3). This should be patient-based, using e.g. the largest lesion of the patient or the total volume of all lesions.

Are there other potential factors/treatment modalities that changed over the 25 year time period, and that are not documented or accounted for in the analysis?

The cutpoints reported in the 1st key point and the discussion (3cm diameter/14cm³ volume) are not analyzed in this study. Or if only lesions with diameter \leq 3cm were included, it is confusing to have a category of 2.5cm to 3.5cm in the analysis. according to table 4, there are lesions with a volume $>$ 14cm³ included.

In contrast to what is described in the statistical methods, gender and volume are not analyzed as predictors in table 5.

The multivariable model described in statistical methods (based on some sort of best subset selection) is not the one presented in table 5. What was the rationale for fitting a multivariable model based on diameter and histology? If only 'significant' predictors from the full multivariable model are reported, this should be changed, and the full model needs to be presented.

What was the rationale for the cutpoints of tumor diameter/volume? Lesion between 1.5 and 2 cm seem to show a very similar pattern/effect as lesions between 2 and 2.5 cm, but do not reach statistical significance in multivariable analysis. Thus, in the abstract and results section, lesions $>$ 2cm are considered risk factor, but in the discussion the authors refer to a cut-off of 1.5cm, i.e. univariable results. This should be reconciled.

Table 4 suggests some missing values for SRS modality. How have missing values been handled in models? Table 5 should indicate the number of patients/lesions/failures evaluable for each model.

Figure 1 indicates 294 lesions with LTF whereas 293 lesions are mentioned in the text

Figure 2 legend states that 339 lesions are under surveillance after 5 years. This does not match with the number at risk.

Overall survival: the number of deaths during FU should be provided

Table 6/7: please provide confidence intervals

Kaplan-Meier plots should indicate censoring

Tumor diameter and year of SRS should be included in table 4

Reviewer #3 (Remarks to the Author):

General comments:

1. Brain metastasis has affected many cancer patients and historically difficulties to be improved for patient survival and quality of life. With advent of early detection and treatments, SRS of solitary or multiple brain metastases become one of the most successful treatments for local tumor control and neurocognitive improvements. By retrospective analysis of a large single institution cohorts of patients, authors of the manuscript have demonstrated their key findings of (1) new brain metastasis ($d < 3$ cm) response differently to standard SRS and (2) melanoma and increased tumor size of ($d > 2$ cm) are main intrinsic factors that shorten the time to local treatment failure (TTF) or poor local control rate.

2. Although the methods and results are well described, I would recommend authors to explore possible clinical impact for their finding, and importantly, provide some radiobiological foundation of their findings. For which, please see some of my following specific comments.

Specific comments:

S1: Lines 91 - 94 : "Despite newer generation systemic therapies with intracranial efficacy (e.g., immunotherapy, tyrosine kinase inhibitors, and antibody drug conjugates), most patients will ultimately

have intracranial disease progression; therefore, local therapies remain crucial in managing brain metastases.” Should be deleted.

S2: Line 155: LTF was defined either radiographically or pathologically after surgery but for Figure 1. Outcomes following SRS for treatment-naïve brain metastasis only list as radiographic findings. Also in the figure 1, 227 LTFs + 66 AREs is 293 which one less than 294 findings. Is it typo or method issue?

S3: Line 194-195 “The TTF ratio is approximately interpretable as the inverse of hazard ratios.” The inverse proportional relationship is right for using Cox proportional hazard model but not necessary for the AFT model. Thus, it is better not make the interpretation.

S4: Lines 201-202 “Model-adjusted differences among the levels of discrete variables in TTF were assessed by Tukey adjusted contrasts.” I am not sure about the term of Tukey adjusted contrasts, do you mean you used Tukey test?

S5: Lines 215-216, “The most common primary tumor type was non-small cell lung cancer (36%), followed by melanoma (21%), breast cancer (12%), and renal cell carcinoma (6%).” “Are inconsistent with % listed in the table 3 or 36%, 20%, 11%, and 6%, please fix it.

S6: Lines 217-218 “The majority of the lesions were treated with Gamma Knife SRS (74%) and the median dose was 20 Gy (range 8-24).” I would say “Seventy-four percent of lesions received Gamma Knife SRS with a mean periphery dose of 20 Gy ranged 8 to 24 Gy.” Please using periphery SRS dose in Table 4. Please comment on Why having the extremely low dose of 8 Gy?

S7: Lines 223-234, 293 agree with 227 LTF + 66 ARE but not agree 294 on the first box in Figure 1. The clinically and radiographically defined LTF of n = 113 in the second box of the bottom row also not agree with your statement of 112 cases. Please correct those typo or changes.

S8: Line 241: “... SRS modality (LINAC v. GK SRS; $P < 0.0001$) significantly influenced TTF (Table 5).” Notice that LINAC SRS has lower effective tumor dose for periphery 20Gy at 80% while GK SRS having 20 Gy at 50%.

S9: Line 248: “>2.5 and £3.5cm (TTF ratio 0.19; 95%CI, 0.12-0.30; $P = 0.004$; Table 5).” while in abstract section (line 62) you have limited to < 3 cm. please change your study limit on lesion size to 3.5 cm

S10: Line 251, The 1- and 2- year local control rate (LCRs)

S11: Lines 252-257, “The 1- and 2- year LCRs for lesions $\leq 0.5\text{cm}$ were 93% and 84% respectively (Figure 2A Volumes in Figure 2B). The 1- and 2-year LCRs for lesions based on diameter ranges were as follows: > 0.5 and $\leq 1\text{cm}$ (91 and 90%); > 1 and $\leq 1.5\text{cm}$ (85 and 79%); > 1.5 and $\leq 2\text{cm}$ (79 and 63%); > 2 and $\leq 2.5\text{cm}$ (70 and 62%); and > 2.5 and $\leq 3.5\text{cm}$ (51 and 32%; Table 6).” Can be concisely stated as

“The 1-and 2-year LCRs for lesions in diameter ranges of 0-0.5, 0.5-1, 1-1.5, 1.5-2, 2-2.5, 2.5-3.5 cm are 93 and 84%, 91 and 90%, 85 and 79%, 79 and 63%, 70 and 62%, and 51 and 32%, respectively.”

S12: Line 257-260, “Amongst tumor histology, melanoma and breast cancers had the lowest 2-year LCRs at 66% and 63% respectively (Table 7; Supplemental Figure S2). Renal cell carcinoma (RCC) and non-small cell lung cancer (NSCLC) had the highest 2-year LCRs at 93% and 83% respectively (Table 7; Supplemental Figure S2).” Please change the lowest to lower and the highest to higher.

S13: Lines 278-279, “Single or multi-fraction SRS can be considered for lesions > 3 to 4 cm diameter (14-30 cm³).” Should be “Hypofraction SRS are recommended for lesions >3 to 5 cm diameter. Authors should aware and add the new ASTRO guideline on “Practical Radiation Oncology_ (2022) 12, 265–282”

S14: Lines 283-284, “This indicates that a multi-fraction technique may not improve the efficacy of SRS for brain metastasis larger than 2cm (4 cm³).” Which may due to use of margins for multiple fractionated SRS for having comparable radio-necrosis and comparable radiobiological doses or Equivalent Dose to 2 Gy per fraction.

S15: Line 290, “We hypothesized that tumor intrinsic properties influence susceptibility of lesions $<3\text{cm}$ to SRS.” Changed to “We hypothesized that tumor and normal tissue intrinsic radiosensitivities influence susceptibility of lesions to SRS.” Traditional linear-quadratic model (or alpha-beta model) may not be appreciated for a large dose per fraction SRS treatments. Please cite the recent publication of a unified multi-activation model by Li et al on “Med Phys. 2021 Apr;48(4):2038-2049”. All of your finding of low LCR for melanoma has been precisely predicted in the study for EQD2 calculated of ~ 35 Gy (24.3 to 41.9 Gy) instead of LQ predicted of EQD2 of 50 Gy for the 20 Gy SRS. This is exciting for matching lab work-based model to clinical observations!

S16: Lines 314-315, “We also utilized perfusion-weighted advanced brain tumor imaging, when available, to distinguish RN from true progression.” You have mentioned using advanced brain tumor imaging (ABTI) on line 157 and if possible, could you provide an example of reduced perfusion for a RN case?

S17: Line 354-456, “Since our results are not influenced by prior or ongoing systemic treatments, these findings indicate that tumor intrinsic properties significantly influence susceptibility to SRS.” Your clinical

findings do agree with latest radiobiological modeling and EQD2 prediction based on lab-observed cell line survival curves.

REVIEWER COMMENTS

Reviewer #1 (Remarks to the Author):

Review of the Manuscript by Ene and colleagues

Introduction: Ene and colleagues investigate the efficacy of stereotactic radiosurgery (SRS) on “smaller” brain metastases (≤ 3 cm in diameter) in patients with no prior systemic therapy. Their study is rooted in the observation that while SRS alone is the treatment of choice for lesions ≤ 3 cm, the responses vary. This study aims to discern the factors behind this variability.

Key results: A significant number of lesions showed MRI findings consistent with local treatment failure (LTF) post-SRS. Notably, tumor size (especially > 2 cm) and melanoma histology were associated with higher LTF rates. Their assertion that brain metastases ≤ 3 cm do not uniformly respond to SRS alone and that guidelines integrating tumor size and histology are required is quite contentious.

RESPONSE

We agree that the optimal treatment approach for brain metastasis ≤ 3 cm remains contentious. However, studies have demonstrated that even though SRS is recommended for lesions ≤ 3 cm, there is still a variable response that is associated with treatment failure in some lesions (Chang EL et al. 2003 PMID: 12925241) and these observations form the basis of our current study that evaluated the factors associated with this variation in response. In addition to our results, which indicate that size and histology may be associated with response in tumors ≤ 3 cm, we have added discussion regarding ongoing/planned prospective studies that will address this question (lines 315-424).

Validity: The manuscript offers valuable insights into SRS's variable effectiveness based on tumor size and histology. However, the recommendation against SRS for certain tumor sizes and histologies seems premature, especially given the well-established efficacy and minimal side effects of SRS.

RESPONSE

We appreciate the feedback. Our intention was not to recommend against SRS for certain tumor sizes or histologies based on our retrospective results, but rather to demonstrate that they support initiating prospective clinical trials (radiation-based and surgery-based) in specific radioresistant histologies. These trials outlined in the discussion section will evaluate the relevance of different SRS modalities (with or without surgery for the 2-3cm radioresistant lesions that have a lower local control rate). See our expanded discussion section and the conclusion sections.

Data are reported during a 25 years time frame with GK + Linac based SRS – prescription differences? I assume this was largely different, but is of utmost importance when interpreting LC data;

RESPONSE:

Thank you for the feedback. We agree with the reviewer's comment and we have now included the prescription differences for LINAC and GK SRS in the methods section (Lines 142-145) and Table 4. In brief, LINAC SRS prescription doses were delivered to 81-95% isodose line while GK SRS prescription doses were delivered to the 50% isodose line. This is our standard protocol at MD Anderson (Chang et al Lancet Oncology 2009, Chang et al Neurosurgery 2003). Although our univariate analysis identified a significant difference in local control between LINAC and GK SRS, this difference was not significant in the multivariate analysis, a finding that is more consistent with prospective studies comparing both modalities (now stated in study limitations lines 435-438). Therefore, we have removed Supplemental figure 1, which depicted a difference in local control between both modalities based on univariate analysis.

the following statement needs to be discussed further:

"Patients with concerning imaging findings who were functionally not fit for further treatment or who chose not to proceed with further treatment were also deemed as having local treatment failure" – problematic as there may have been several cases of pseudoprogression; outcome of these patients?

RESPONSE

Thank you for highlighting this. We agree that distinguishing true tumor progression from pseudoprogression without tissue has historically been problematic. We agree that we cannot rule out pseudoprogression in all the patients with 'concerning imaging findings'. In these cases, we have relied on outcomes from standard imaging criteria (and perfusion data from advanced MRI if available) AND clinical decision-making as the rationale for deeming the cases local treatment failure. For instance, if a lesion is deemed to have progressed based on MRI characteristics and the patient is offered palliative treatment (ex. WBRT) to address this progression, this is categorized as local treatment failure based on the clinical decision to offer palliative treatment (independent of if they choose to move forward with palliative treatment or they are not functionally fit for treatment). There were 26 patients with a total of 36 lesions (2% of all treated lesions) with concerning imaging findings of the treated brain metastasis, who were functionally not fit for further treatment or who chose not to proceed with further treatment. Perfusion data was available for 3 out of the 36 lesions, all lesions were consistent with a viable tumor signature and therefore likely tumor progression. 17/26 patients went on to hospice care after declining clinical intervention indicating that this cohort were likely a poor functional status from systemic progression in addition to the concern for intracranial progression of the SRS treated lesion. We have expanded on this in the methods section lines 158-159 with a statement 'pseudoprogression cannot be ruled out entirely in patients who chose not to proceed with treatment'. We have also included the outcomes of patients in this category in the results section under outcomes in lines 236-242 and discussed as a limitation in lines 443-445.

Significance: While the article points out the potential radioresistance of melanomas and the size-dependent response to SRS, these findings are not novel. However, the data's significance lies in its comprehensive analysis from a size perspective, which is often overlooked in pooled data.

RESPONSE

Thank you for the feedback. We agree that the radioresistance and size dependent response are not novel but our detailed systematic and comprehensive lesion-based analysis from a treatment naïve cohort provides an in-depth insight into the basis for variability in the response for lesions <3cm. Our results provide a framework for evaluating outcomes of ongoing prospective studies and helps with the planning of future and histology-based prospective studies as we have now outlined in the discussion section.

Data and methodology: The retrospective analysis of a large single-institution cohort over 25 years is a significant strength. However, there are inconsistencies, which may still arise from misallocation/misinterpretation.

RESPONSE

We appreciate the opportunity to address the inconsistencies raised. We have now addressed the issues brought up during the review process especially regarding over-interpretation of results from a retrospective study. We have removed the overstatement regarding SRS efficacy for lesions >2cm and proposed ways to enhance the local control of SRS for this group of lesions such as fractionated courses, synergy with surgery, immune checkpoint inhibitors or radiosensitizers. Please see an expanded discussion section with literature review.

Analytical approach: While the statistical approach, particularly concerning size, is rigorous and differentiates this study from others, some results contradict established literature, raising questions about potential errors or biases. In the methods section, the paragraph could be shortened and simplified a bit.

RESPONSE

Thank you for the comments. We have expanded our study limitations section. We agree that one limitation of a retrospective study is the selection bias that could influence the results. The main findings from the study, regarding the radioresistance of melanoma are consistent with published literature. One consideration that may contribute to some other differences noted in outcomes between our study and others may lie in the fact that our cohort is exclusively treatment-naïve while other studies pool lesions from patients receiving a diverse array of local or systemic treatments that could influence outcomes (now stated in limitations section lines 439-442).

Another difference noted in local control between LINAC and GK SRS was based on univariate analysis and it was not significant in the multivariate analysis (which is consistent with published and cited prospective studies. We have added this to the study limitations section in Lines 435-438. To avoid this confusion, we have removed

supplemental figure 1 which showed a difference in local control rates between LINAC and GK SRS since this did not reach significance in the multivariate analysis. The study was a large cohort with significant statistical input and we have only included the details that are needed for reproducibility, with simplification of description provided when able.

Suggested improvements:

- please correct line 96 regarding trial EORTC 22952-26001.

RESPONSE

Thank you for this comment. We have corrected in the text as suggested (line 96),

- please clarify: "The majority of the lesions were treated with Gamma Knife SRS (74%) and the median dose was 20 Gy (range 8-24)." – means there were hypofractionated courses included as well?

RESPONSE

Thanks for pointing this out and we agree that the 8Gy is unusual. There were no hypofractionated course but the range is 8-24Gy because one calvarial metastasis received 8Gy. This is now explicitly stated in lines 217-218.

- Please discuss the main flaw: Radiation necrosis (RN) based on 230 pathology after surgery occurred in 28 lesions (2% of all treated lesions) – far too low compared to expectations when delivering ablative doses

RESPONSE

Thanks for pointing this out. 2% (28 out of 1733 treated lesions) only covers pathology that had 'pure' radiation necrosis without viable tumor at surgery. 3% (48 out of 1733 treated lesions) had mixed pathology (Viable tumor AND radiation necrosis) therefore there were more lesions with radiation necrosis, but these lesions were deemed local treatment failure because there was viable tumor on pathology and radiographically fit the criteria for local treatment failure (See table 2 for criteria). Additionally, 1% (18 out of 1733 treated lesions) fit clinical and radiographic criteria for an adverse radiation effect that resolved with steroids or avastin (See table 2 for criteria). Therefore, the overall incidence of radiation necrosis (pure and mixed) is higher at 5% (48+28+18)/1733. These findings are consistent with previous reports following SRS (see references below). Additionally, over 85% of the lesions in this study were <2cm in diameter, where SRS is associated with a lower incidence of radiation necrosis.

To provide clarity, the results section (lines 228-235) is now updated to reflect this information with the overall incidence of combined RN now stated.

Figure 1 is now also updated to state 'pure radiation necrosis without viable tumor surgery'.

The references below are also now included in the discussion (lines 410-412).

Kohutek, Z. A. *et al.* Long-term risk of radionecrosis and imaging changes after stereotactic radiosurgery for brain metastases. *J Neurooncol* **125**, 149-156 (2015). <https://doi.org/10.1007/s11060-015-1881-3>

Minniti, G. *et al.* Stereotactic radiosurgery for brain metastases: analysis of outcome and risk of brain radionecrosis. *Radiat Oncol* **6**, 48 (2011). <https://doi.org/10.1186/1748-717X-6-48>

Sneed, P. K. *et al.* Adverse radiation effect after stereotactic radiosurgery for brain metastases: incidence, time course, and risk factors. *J Neurosurg* **123**, 373-386 (2015).

- Information on margins used, Dx values and when hypofractionation was used are largely missing

RESPONSE

Fractionation was not used in these series. As previously stated, the 8Gy dose was a single dose to a calvarial lesion (Line 217). Prescription doses are now shown in table 1. In Line 142-144 of the methods section, we have also now stated that LINAC prescription dose was delivered to 80-95% isodose line while GK was delivered uniformly to the 50% isodose line as we have previously described based on our experience at MD Anderson (relevant papers from MD Anderson now cited).

- Please discuss the alternative scenario that hypofractionation might overcome existing limitations – as even surgery + SRS is associated with high failure rates, specifically when the cavity volume was large

RESPONSE

Thank you for this comment. In lines 315-338, we have now extensively discussed the role fractionation may play in overcoming existing limitations for achieving local control in radioresistant lesions.

Clarity and context: The article is generally clear, but it could benefit from a more on depth discussion contextualizing its findings within the larger body of literature. It is vital to recognize the novel contributions without overstating them.

RESPONSE

We have now expanded our discussion section to include ongoing and proposed prospective clinical trials that may help evaluate the questions raised in our retrospective analysis. We recognize and state that our findings need to be validated in these clinical trials and our findings may also provide the framework for future histology-specific clinical trials that evaluate different SRS modalities or combination strategies with surgery or immune checkpoint inhibitors or radiosensitizers.

Overall: The manuscript brings valuable insights, especially with its rigorous statistical analysis concerning tumor size. However, it is crucial to reconcile findings with existing literature and not draw broad conclusions without the robustness of randomized trials. Specifically as due to Kjellberg's observation, prescription is lowered when size grows as RN shall be avoided - which in turn explains lower control rates if the RN rate is low.

RESPONSE

We appreciate the feedback. Based on Kjellberg's findings, it is our practice at MD Anderson to administer lower doses for larger tumors or consider fractionated SRS for larger lesions >3cm when surgery is not feasible. We have now stated this in line 324 with Kjellberg's work cited. It is still unclear if fractionated SRS will achieve similar or better local control rates and lower incidence of RN) compared with single fraction SRS particularly for radioresistant lesions in the size category identified in this retrospective study. We now state that ongoing/proposed prospective clinical trials will address this.

Reviewer #2 (Remarks to the Author):

This is the biostatistical review. This is a retrospective analysis of data from a single center including patients from 25 years.

The authors perform a lesion-based analysis with endpoint time to local failure. They fit an AFT model with log-logistic distribution accounting for clustered data due to multiple lesions per patients. While the overall approach is fine, there are some discrepancies between described methods and reported results, and some points need clarification.

COMMENT #1: Endpoint TTF: the authors state that FU was censored at last FU, salvage surgical resection, or ARE. But they also state 'For patients who underwent salvage surgical resection, pathology showing viable tumor was deemed LTF'. Please clarify.

RESPONSE: This was a typographical error as patients receiving salvage surgical resection were deemed LTF. This error is now corrected in lines 171-172.

COMMENT #2: How were the other lesions of the same patient treated in case one lesion showed local failure?

RESPONSE: The accelerated failure time models clustered on patient to account for multiple lesions per patient. Kaplan-Meier models treated each tumor as an independent sample. Kaplan-Meier figures are provided as illustrations; inference derived only from the accelerated failure time models. Accordingly, line 179-181 has been revised to add this clarification:

TTF was summarized by Kaplan-Meier method for discrete variables, with each lesion treated as an independent sample (within-patient clustering was subsequently accommodated within appropriate models).

COMMENT #3: Endpoint OS: I do not think a lesion-based analysis for OS is sensible (suppl fig 3). This should be patient-based, using e.g. the largest lesion of the patient or the total volume of all lesions.

RESPONSE: We agree that using overall survival as an endpoint does not make sense, therefore we have removed supplemental figure 3. This is an important point, because overall survival in these patients is not driven by the brain metastasis, so the data is not helpful.

Are there other potential factors/treatment modalities that changed over the 25 year time period, and that are not documented or accounted for in the analysis?

RESPONSE: Yes, since multivariate analysis showed that the year of SRS significantly influenced outcomes, we agree that the era of treatment most likely influenced outcomes. We have now added this to the study limitations in lines 438-442 and expanded on possible reasons for this including the availability/use of more effective brain penetrant chemotherapies and the use of immunotherapy after SRS was administered to treatment naïve lesions.

COMMENT #4: The cutpoints reported in the 1st key point and the discussion (3cm diameter/14cm³ volume) are not analyzed in this study. Or if only lesions with diameter ≤ 3 cm were included, it is confusing to have a category of 2.5cm to 3.5cm in the analysis. according to table 4, there are lesions with a volume >14 cm³ included.

RESPONSE: This study focused on diameter associations (a prior iteration had focused on volume, which led to some confusion). Diameter intervals are at 0.5 cm intervals between 0 and 3 cm; based upon discussion of this comment, we have excluded data from 3 patients with lesions exceeding 3 cm. To assist interpretation by readers more familiar with volumes, a corresponding volume interval scale has been introduced in table 4 and we removed the volume chart in figure 2B to avoid confusion. Also, figure 2 is now updated to reflect inclusion of only lesions with diameter ≤ 3 cm. All tables and figures also reflect the exclusion of lesions >3 cm.

COMMENT #5: In contrast to what is described in the statistical methods, gender and volume are not analyzed as predictors in table 5. The multivariable model described in statistical methods (based on some sort of best subset selection) is not the one presented in table 5. What was the rationale for fitting a multivariable model based on diameter and histology? If only 'significant' predictors from the full multivariable model are reported, this should be changed, and the full model needs to be presented.

RESPONSE: Thank you for bringing this to our attention. The methods with regard to multivariable modeling had been incompletely revised following a change which replaced modeling with relation to volume to modeling with relation to diameter. Under the Univariate Analysis in Table 5, summaries for Gender and Post SRS WBRT have now been added. We have also reported the percent local control based on gender in supplemental figure S3.

The exhaustive variable selection in the multivariable model is described in lines 191-195, and has been revised to state that the variables explored all combinations of “age, year of treatment, primary tumor histology, discrete diameter, LINAC vs. Gamma knife SRS, KPS, SRS dose, and Post-SRS WBRT status”.

Likewise, the optimal multivariable model described in lines 195-197 has been revised to state, “assessed the association between TTF and the variables of age, year of treatment, primary tumor histology, discrete diameter, and Post-SRS WBRT status.”

Since the focus of this study was the association between time to treatment failure and tumor diameter, the model associations with age, year of treatment, primary tumor histology, and Post-SRS WBRT status were considered as controlled-for covariates, and thus not necessary to report due to being only incidental to the association of interest. Association with primary tumor histology had been included in the summary due to investigator interest. Summaries for age, year of treatment, and Post-SRS WBRT status have now been added to Table 5 to address this comment.

COMMENT #6: What was the rationale for the cutpoints of tumor diameter/volume? Lesion between 1.5 and 2 cm seem to show a very similar pattern/effect as lesions between 2 and 2.5 cm, but do not reach statistical significance in multivariable analysis. Thus, in the abstract and results section, lesions >2cm are considered risk factor, but in the discussion the authors refer to a cut-off of 1.5cm, i.e. univariable results. This should be reconciled.

RESPONSE: Diameter cutpoints were at 0.5 cm intervals between 0 and 3.0. The 0.5 cm interval was chosen as clinically relevant and interpretable. The abstract and results sections are now reconciled with emphasis on multivariate analysis showing a cutoff of 2cm.

COMMENT #7: Table 4 suggests some missing values for SRS modality. How have missing values been handled in models? Table 5 should indicate the number of patients/lesions/failures evaluable for each model.

RESPONSE: Those counts have been added to Table 5. With reference to the prior draft, data for 3 patients with lesions exceeding 3 cm has been excluded, therefore, these numbers would differ. KPS data was available for a subset of 1076 patients with 1702 lesions and 223 failures; other variables had complete data.

COMMENT #8: Figure 1 indicates 294 lesions with LTF whereas 293 lesions are mentioned in the text

RESPONSE: Figure 1 was corrected it should be 291, further updating to reflect exclusion of 3 patients with lesion diameter exceeding 3cm.

COMMENT #9: Figure 2 legend states that 339 lesions are under surveillance after 5 years. This does not match with the number at risk.

RESPONSE: The Figure 2 legend was corrected to indicate that 5 failures occurred after 5 years.

COMMENT #10: Overall survival: the number of deaths during FU should be provided

RESPONSE: This information has been added to Table 3.

COMMENT #11: Table 6/7: please provide confidence intervals

RESPONSE: These are now added.

COMMENT #12: Kaplan-Meier plots should indicate censoring

RESPONSE: Censoring marks have been added to Kaplan-Meier plots.

COMMENT #13: Tumor diameter and year of SRS should be included in table 4

RESPONSE: Summaries for Tumor diameter (and corresponding volumes) and year of SRS have been added to Table 4.

Reviewer #3 (Remarks to the Author):

General comments:

Brain metastasis has affected many cancer patients and historically difficulties to be improved for patient survival and quality of life. With advent of early detection and treatments, SRS of solitary or multiple brain metastases become one of the most successful treatments for local tumor control and neurocognitive improvements. By retrospective analysis of a large single institution cohorts of patients, authors of the manuscript have demonstrated their key findings of (1) new brain metastasis ($d < 3$ cm) response differently to standard SRS and (2) melanoma and increased tumor size of ($d > 2$ cm) are main intrinsic factors that shorten the time to local treatment failure (TTF) or poor local control rate.

Although the methods and results are well described, I would recommend authors to explore possible clinical impact for their finding, and importantly, provide some radiobiological foundation of their findings.

RESPONSE

We have now included an in-depth discussion (lines 315-338) about the radiobiological implications of our findings, especially how they corroborate recent preclinical models for predicting the EQD2 for SRS-treated brain metastasis of melanoma. Importantly, these preclinical models allow us to propose a framework for designing prospective clinical trials to assess how fractionation, especially for radioresistant lesions, could enhance local control while minimizing radiation necrosis. We also discuss how histology-specific genetic alterations could influence susceptibility to radiosensitizers and immunotherapy (lines 369-392)

For which, please see some of my following specific comments.

Specific comments:

S1: Lines 91 - 94 : “Despite newer generation systemic therapies with intracranial efficacy (e.g., immunotherapy, tyrosine kinase inhibitors, and antibody drug conjugates), most patients will ultimately have intracranial disease progression; therefore, local therapies remain crucial in managing brain metastases.” Should be deleted.

RESPONSE

Thank you for your valuable feedback and constructive comments on our manuscript. This has been deleted from the text.

S2: Line 155: LTF was defined either radiographically or pathologically after surgery but for Figure 1. Outcomes following SRS for treatment-naïve brain metastasis only list as radiographic findings. Also in the figure 1, 227 LTFs + 66 AREs is 293 which one less than 294 findings. Is it typo or method issue?

RESPONSE

Thanks for bringing this to our attention. We believe the reviewer is inquiring about the first box in figure 1 where it states radiographic findings concerning for local treatment failure. If so, then yes, this only shows that the concern for LTF started with MRI findings but true LTF was determined later with additional pathology or clinical decision to treat. This was a typographical error and is now updated based on the exclusion of 3 lesions that were >3cm.

The corrections are now made to figure 1: 291 lesions total, 227 LTF and 64 ARE. There were 115 cases of surgery confirmed LTF and 112 cases of clinical or radiographic LTF.

S3: Line 194-195 “The TTF ratio is approximately interpretable as the inverse of hazard ratios.” The inverse proportional relationship is right for using Cox proportional hazard model but not necessary for the AFT model. Thus, it is better not make the interpretation.

RESPONSE

Since many investigators are accustomed to interpreting hazard ratios, we feel that it is helpful to provide this interpretation to provide some context of directionality of the TTF ratios. We agree that this is not a 1-to-1 transformation. Including the “approximately” provides a sense of this limitation for non-statisticians.

S4: Lines 201-202 “Model-adjusted differences among the levels of discrete variables in TTF were assessed by Tukey adjusted contrasts.” I am not sure about the term of Tukey adjusted contrasts, do you mean you used Tukey test?

RESPONSE:

Yes, the Tukey adjustment to model contrast p-values is sometimes referred to as a Tukey test. We have changed to Tukey Test in Line 198.

S5: Lines 215-216, “The most common primary tumor type was non-small cell lung cancer (36%), followed by melanoma (21%), breast cancer (12%), and renal cell carcinoma (6%). “Are inconsistent with % listed in the table 3 or 36%, 20%, 11%, and 6%, please fix it.

RESPONSE:

The types of primary tumors are now consistent with table 3. 36%, 21%, 11%, and 6%. Lines 212-214.

S6: Lines 217-218 “The majority of the lesions were treated with Gamma Knife SRS (74%) and the median dose was 20 Gy (range 8-24).” I would say “Seventy-four percent of lesions received Gamma Knife SRS with a mean periphery dose of 20 Gy ranged 8 to 24 Gy.” Please using periphery SRS dose in Table 4. Please comment on Why having the extremely low dose of 8 Gy?

RESPONSE

Thank you for the recommendation. This has now been changed accordingly. We have also indicated that the low 8Gy dose was for one case of a calvarial metastasis and we state that there were no fractionated cases (line 217-218). We have now added periphery SRS dose to table 4.

S7: Lines 223-234, 293 agree with 227 LTF + 66 ARE but not agree 294 on the first box in Figure 1. The clinically and radiographically defined LTF of n = 113 in the second box of the bottom row also not agree with your statement of 112 cases. Please correct those typo or changes.

RESPONSE

This was a typo and is now updated based on the exclusion of 3 lesions that were >3cm.

The corrections are now made to figure 1. 291 lesions total. 227 LTF and 64 ARE. There were 115 cases of surgery confirmed LTF and 112 cases of clinical or radiographic LTF.

S8: Line 241:”... SRS modality (LINAC v. GK SRS; P<0.0001) significantly influenced TTF (Table 5).” Notice that LINAC SRS has lower effective tumor dose for periphery 20Gy at 80% while GK SRS having 20 Gy at 50%.

RESPONSE

This statement was based on univariate analysis but it did not reach significance in the multivariate analysis. This is now stated under univariate and multivariate section of results. We have removed supplemental figure 1 as multivariate analysis did not show a difference between LINAC and GK SRS, which is consistent with what prospective clinical trials have shown.

In the methods, we have also clarified how the prescription doses were administered: LINAC SRS prescription doses were delivered to the 80-95% isodose line while GK SRS prescription doses were delivered to the 50% isodose line” (Line 142-145).

S9: Line 248: “>2.5 and <3.5cm (TTF ratio 0.19; 95%CI, 0.12-0.30; P =0.004; Table 5).”. while in abstract section (line 62) you have limited to < 3 cm. please change your study limit on lesion size to 3.5 cm

RESPONSE

We have now changed our study limit to lesions size to <3cm since this is the group with a variable response. Therefore, we have excluded 3 lesions and updated the statistical analysis accordingly.

S10: Line 251, The 1- and 2- year local control rate (LCRs)

RESPONSE

LCR is now spelled out completely. Line 258.

S11: Lines 252-257, “The 1- and 2- year LCRs for lesions ≤ 0.5 cm were 93% and 84% respectively (Figure 2A Volumes in Figure 2B). The 1- and 2-year LCRs for lesions based on diameter ranges were as follows: > 0.5 and ≤ 1 cm (91 and 90%); > 1 and ≤ 1.5 cm (85 and 79%); > 1.5 and ≤ 2 cm (79 and 63%); > 2 and ≤ 2.5 cm (70 and 62%); and > 2.5 and ≤ 3.5 cm (51 and 32%; Table 6).” Can be concisely stated as “The 1-and 2-year LCRs for lesions in diameter ranges of 0-0.5, 0.5-1, 1-1.5, 1.5-2, 2-2.5, 2.5-3.5 cm are 93 and 84%, 91 and 90%, 85 and 79%, 79 and 63%, 70 and 62%, and 51 and 32%, respectively.”

RESPONSE

Thank you for the feedback. We have made this change to make the statement more concise. Line 261-263.

S12: Line 257-260, “Amongst tumor histology, melanoma and breast cancers had the lowest 2-year LCRs at 66% and 63% respectively (Table 7; Supplemental Figure S2). Renal cell carcinoma (RCC) and non-small cell lung cancer (NSCLC) had the highest 2-year LCRs at 93% and 83% respectively (Table 7; Supplemental Figure S2).” Please change the lowest to lower and the highest to higher.

RESPONSE

This change is now complete. Lines 263-265.

S13: Lines 278-279, “Single or multi-fraction SRS can be considered for lesions > 3 to 4 cm diameter (14-30 cm³).” Should be “Hypofraction SRS are recommended for lesions >3 to 5 cm diameter. Authors should aware and add the new ASTRO guideline on “Practical Radiation Oncology_ (2022) 12, 265–282”

RESPONSE

Thank you for shedding the light to the new ASTRO guidelines. We have made the necessary adjustments in text as suggested in Lines 285-286 with the recommended references.

S14: Lines 283-284, “This indicates that a multi-fraction technique may not improve the

efficacy of SRS for brain metastasis larger than 2cm (4 cm³)." Which may due to use of margins for multiple fractionated SRS for having comparable radio-necrosis and comparable radiobiological doses or Equivalent Dose to 2 Gy per fraction.

RESPONSE

We have changed this statement based on most recent guidelines and evidence regarding SRS hyperfractionation. We now discuss how EQD2 could allow delivery of the same dose of SRS to melanoma (line 315-319). We also now expand on the ongoing clinical efforts to determine the efficacy of multifraction SRS for local control and risk of radiation necrosis in the discussions section.

S15: Line 290, "We hypothesized that tumor intrinsic properties influence susceptibility of lesions <3cm to SRS." Changed to "We hypothesized that tumor and normal tissue intrinsic radiosensitivities influence susceptibility of lesions to SRS." Traditional linear-quadratic model (or alpha-beta model) may not be appreciated for a large dose per fraction SRS treatments. Please cite the recent publication of a unified multi-activation model by Li et al on "Med Phys. 2021 Apr;48(4):2038-2049". All of your finding of low LCR for melanoma has been precisely predicted in the study for EQD2 calculated of ~35 Gy (24.3 to 41.9 Gy) instead of LQ predicted of EQD2 of 50 Gy for the 20 Gy SRS. This is exciting for matching lab work-based model to clinical observations!

RESPONSE

This is indeed exciting to see a preclinical lab-based model matching clinical observation. We have added this reference to Line 315-319. Our hope is that we can begin to uncover strategies to sensitive the resistant lesions to SRS or modify treatment plan based on histology.

S16: Lines 314-315, "We also utilized perfusion-weighted advanced brain tumor imaging, when available, to distinguish RN from true progression." You have mentioned using advanced brain tumor imaging (ABTI) on line 157 and if possible, could you provide an example of reduced perfusion for a RN case?

RESPONSE

We have now added a case of RN with low relative cerebral blood flow as supplemental figure S5.

S17: Line 354-456, "Since our results are not influenced by prior or ongoing systemic treatments, these findings indicate that tumor intrinsic properties significantly influence susceptibility to SRS." Your clinical findings do agree with latest radiobiological modeling and EQD2 prediction based on lab-observed cell line survival curves.

RESPONSE

Thank you for pointing this out. We have also cited the EQD2 predication paper and hope that these 2 corroborative findings will raise questions about modifying treatment plans based on tumor histology.

REVIEWER COMMENTS

Reviewer #1 (Remarks to the Author):

The authors have made substantial changes to improve the overall quality of the manuscript, I'm fine with the current version - congratulations.

Reviewer #2 (Remarks to the Author):

Thank you, the authors have addressed most of my points, there are some few points left and some new ones:

In regard to my previous COMMENT #2: I just wanted to clarify that if one lesion of a patient had a local failure how were the other lesions of that patient handled in the further follow-up for the analysis?

Line 438/439: The authors write that SRS modality was not significant in the multivariable model. But SRS modality is not part of the final multivariable model and AIC based variable selection is not the same as testing a variable.

Since OS was removed altogether from the manuscript, the authors could remove number of deaths from table 3 and the last sentence on OS in the methods section.

There is now a new treatment factor included in the model: Post-SRS WBRT. Is that a post-baseline factor? And if so, was it treated as time-dependent factor?

Line 183/184: sentence on considering Cox regression but using AFT was shortened and is now not easy to understand. Maybe to remove the sentence entirely.

Line 254/255 was probably not updated, it still reads '>2.5 and <=3.5cm (TTF ratio 0.19; 95%CI, 0.12-0.30; P=0.004; Table 5)'

Table 5: the reference category for sex should be indicated

Reviewer #3 (Remarks to the Author):

Authors have responded and corrected all of my comments and no further comment or suggestion for the revised manuscript.

Response to Reviewer comments

Reviewer #1 (Remarks to the Author):

The authors have made substantial changes to improve the overall quality of the manuscript, I'm fine with the current version - congratulations.

Thank you for your feedback, it improved the manuscript significantly.

Reviewer #2 (Remarks to the Author):

Thank you, the authors have addressed most of my points, there are some few points left and some new ones:

In regard to my previous COMMENT #2: I just wanted to clarify that if one lesion of a patient had a local failure how were the other lesions of that patient handled in the further follow-up for the analysis?

All of the accelerated failure time models clustered on patient, so for patients with multiple lesions there would effectively be an averaged within-patient trend over those lesions. For the multivariable model, this included relevant adjustments for specific tumor diameter.

Line 438/439: The authors write that SRS modality was not significant in the multivariable model. But SRS modality is not part of the final multivariable model and AIC based variable selection is not the same as testing a variable.

Agreed, this text is residual from an earlier iteration, and has been removed.

Since OS was removed altogether from the manuscript, the authors could remove number of deaths from table 3 and the last sentence on OS in the methods section.

Agreed. Death is now removed from table 3 and overall survival is also removed from the methods section (lines 187).

There is now a new treatment factor included in the model: Post-SRS WBRT. Is that a post-baseline factor? And if so, was it treated as time-dependent factor?

Thanks for bringing this to our attention. We have taken several steps to address this important issue.

1. Since additional radiation therapy to the brain via whole brain radiotherapy (WBRT) may alter the outcomes following SRS, patients are now censored at time of post-SRS WBRT if local failure of the SRS treated lesion had not occurred when WBRT was given. This is stated explicitly in methods section line 193-195 and in the local control rates section of results (Line 291-292).
2. Additionally, we have now omitted WBRT as a covariate from the model given this potential impact of WBRT on local failure (Line 207).
3. Following censor of post-SRS treated patients receiving WBRT, the multivariate analysis showed that lesions >1.5cm in diameter now had significantly shorter time to treatment failure compared to lesions <1.5cm diameter. There was no significant impact on the findings related to tumor histology. The manuscript is now updated to reflect this including the conclusion (lines 525-527).
4. On the side, we also performed an analysis where we completely excluded patients receiving WBRT after SRS (10% of cohort) and this yielded nearly identical outcomes.

Line 183/184: sentence on considering Cox regression but using AFT was shortened and is now not easy to understand. Maybe to remove the sentence entirely.

Agreed. It was shortened to simplify the description. However, we have now deleted it completely to avoid confusion.

Line 254/255 was probably not updated, it still reads '>2.5 and <=3.5cm (TTF ratio 0.19; 95%CI, 0.12-0.30; P=0.004; Table 5)'

This is now corrected.

Table 5: the reference category for sex should be indicated

Agreed. Male v. Female is now indicated in Table 5.

Reviewer #3 (Remarks to the Author):

Authors have responded and corrected all my comments and no further comment or suggestion for the revised manuscript.

Appreciate the feedback which improved the manuscript significantly.

REVIEWERS' COMMENTS

Reviewer #2 (Remarks to the Author):

I have no further comments.